# Three-Dimensional Magnetotelluric Inversion for Triaxial Anisotropic Medium in Data Space

**Jingtao Xie [1], Hongzhu Cai [1,2,*](D), Xiangyun Hu [1,2](D), Shixin Han [1] and Minghong Liu [1]**

1    Institute of Geophysics and Geomatics, China University of Geosciences, Wuhan 430074, China; jtxie@cug.edu.cn (J.X.); xyhu@cug.edu.cn (X.H.); hsx@cug.edu.cn (S.H.); liuminghong@cug.edu.cn (M.L.)
2    State Key Laboratory of Geological Processes and Mineral Resources, China University of Geosciences, Wuhan 430074, China
*    Correspondence: caihongzhu@hotmail.com or caihz@cug.edu.cn

**Abstract:** The interpretation of three-dimensional (3-D) magnetotelluric (MT) data is usually based on the isotropic assumption of the subsurface structures, and this assumption could lead to erroneous interpretation in the area with considerable electrical anisotropy. Although arbitrary anisotropy is much closer to the ground truth, it is generally more challenging to recover full anisotropy parameters from 3-D inversion. In this paper, we present a 3-D triaxial anisotropic inversion framework using the edge-based finite element method with a tetrahedral mesh. The 3-D inverse problem is solved by the Gauss-Newton (GN) method which shows fast convergence behavior. The computation cost of the data-space method depends on the size of data, which is usually smaller than the size of model; therefore, we transform the inversion algorithm from the model space to the data space for memory efficiency. We validate the effectiveness and applicability of the developed algorithm using several synthetic model studies.

**Keywords:** magnetotelluric; triaxial anisotropic; data-space; 3-D inversion; finite element





## 1. Introduction

With the fast development of computer hardware and numerical algorithm, 3-D magnetotelluric (MT) inversion has been widely used in many applications, such as mineral exploration [1,2], deep earth structure imaging [3–5], and geothermal exploration [6,7].

At present, the interpretation of magnetotelluric data is generally based on the isotropy assumption of the Earth's geoelectric structure, but numerous studies have revealed the prevalence of electrical anisotropy in the Earth [8–10]. This electric anisotropy can affect the magnetotelluric response [11,12]. If the isotropic inversion technique was adopted to invert the data affected by electrical anisotropy, it could produce artifacts and even lead to erroneous interpretation [13,14]. Although the arbitrary anisotropy assumption is much closer to the ground truth, it needs to consider more model parameters compared to the isotropic inversion. Moreover, it is more challenging to solve the arbitrary anisotropy inversion due to the strong non-uniqueness [15].

The current research of anisotropic MT inversion mainly focuses on two-dimension (2-D) inversion and 3-D triaxial inversion in the model space. Li et al. (2003) implemented 2-D anisotropic inversion using the Gauss-Newton method [16]. Pek et al. (2011) used the nonlinear conjugate gradient (NLCG) method to solve 2-D arbitrary anisotropic inversion, and they found that it is challenging to reconstruct the anisotropic dipping angle without additional a priori information [17]. Yu et al. (2022) reported 2-D arbitrary anisotropic MT inversion using the limited-memory quasi-Newton (Q-N) method [18]. For 3-D anisotropic inversion, Cao et al. (2018) and Wang et al. (2021) realized 3-D magnetotelluric inversion with triaxial anisotropy using the limited-memory Broyden–Fletcher–Goldfarb–Shanno (L-BFGS) method [15,19]. Kong et al. (2022) reported 3-D arbitrary anisotropic MT inversion using the NLCG method [20]. These published results are based on the model-space

algorithm, and the memory consumption depends on the size of model. Considering the number of model parameters is relatively large for 3-D inversion, the iterative solver is usually used to solve the model update in each inversion iteration.

We develop a 3-D triaxial anisotropic inversion scheme using the data space method, and the Gauss–Newton method is used to solve this inversion problem. To accurately simulate the rugged topography and complex geoelectric structures, the unstructured tetrahedral mesh is used to discretize the computational domain. For simplicity, we now only consider three principal resistivities. The anisotropic strike can be determined by the phase tensor technique and induction arrows in some cases [21,22], and also by considering the known geological information. At present, it is still challenging to resolve the anisotropic dip and slant using the current inversion strategy. The anisotropic inversion algorithm is transformed from the model space to the data space to reduce the memory requirement [23–26]. Moreover, the direct solver is used to solve the normal equation in each inversion iteration. Compared to the iterative solver, the direct solver can improve the stability of the inversion process.

In this paper, we first introduce the anisotropic forward modeling with a secondary field formulation. Following this, we describe the triaxial anisotropic inversion based on the Gauss–Newton optimization in the data space. Finally, we demonstrate the effectiveness and applicability of the developed algorithm using several synthetic models.

## 2. Basic Theory

### 2.1. 3-D Magnetotelluric Forward Modeling with a Secondary Field Formulation

We start from Maxwell's equations and assume the time dependence term of $e^{i\omega t}$. By ignoring the displacement current, we can obtain the diffusion equation as follows:

$$\nabla \times \nabla \times \mathbf{E}^s + i\omega\mu_0 \hat{\boldsymbol{\sigma}} \mathbf{E}^s = -i\omega\mu_0 \left( \hat{\boldsymbol{\sigma}} - \sigma_p \right) \mathbf{E}^p, \tag{1}$$

where $\mathbf{E}^s$ and $\mathbf{E}^p$ are the secondary field and the primary field, respectively; $\omega$ is the angular frequency; $\mu_0$ is the free space magnetic permeability; $\sigma_p$ is the background conductivity, and $\hat{\boldsymbol{\sigma}}$ is the anisotropic conductivity tensor, described as follows [27]:

$$\hat{\boldsymbol{\sigma}} = \mathbf{R}_z(-\alpha_S)\mathbf{R}_x(-\alpha_D)\mathbf{R}_z(-\alpha_L) \begin{bmatrix} \sigma_x & & \\ & \sigma_y & \\ & & \sigma_z \end{bmatrix} \mathbf{R}_z(\alpha_S)\mathbf{R}_x(\alpha_D)\mathbf{R}_z(\alpha_L), \tag{2}$$

where $\sigma_x$, $\sigma_y$, and $\sigma_z$ are three principal conductivities in the X, Y, and Z directions, respectively; $\alpha_S$, $\alpha_D$, and $\alpha_L$ are the anisotropic strike, dip, and slant, respectively; and $\mathbf{R}_z(\alpha_S)$, $\mathbf{R}_x(\alpha_D)$, and $\mathbf{R}_z(\alpha_L)$ represent the rotations around the Z axis and X axis through the angle $\alpha_S$, $\alpha_D$ and $\alpha_L$, respectively.

The large sparse system of linear equations can be obtained by applying finite element analysis:

$$\mathbf{AE}^s = \mathbf{b}. \tag{3}$$

The Dirichlet boundary condition is adopted for solving the above equation by assuming the secondary electrical field vanishes on the boundary of the modeling domain. We use the tetrahedral mesh to discretize the computational domain. Compared to the total field formulation, one has to select proper background conductivity when simulating models with complex topography using the secondary field formulation. We solve Equation (3) using the parallel direct solver MKL Paradiso [28]. Considering the independence of frequency, we further parallelize the forward modeling algorithm over the frequencies to speed up the computation.

### 2.2. Data-Space Inversion Theory for Triaxial Anisotropic Medium

For the triaxial anisotropic inversion, three anisotropic rotation angles are constant, and the objective functional can be written as:

$$\varphi(\mathbf{m}) = \varphi_d(\mathbf{m}) + \beta_x \varphi_x(\mathbf{m}_x) + \beta_y \varphi_y(\mathbf{m}_y) + \beta_z \varphi_z(\mathbf{m}_z). \tag{4}$$

where $\varphi_d(\mathbf{m})$ is the data misfit; $\varphi_x(\mathbf{m}_x)$, $\varphi_y(\mathbf{m}_y)$, and $\varphi_z(\mathbf{m}_z)$ are the model regularization term in the X, Y, and Z directions, respectively; $\mathbf{m} = \begin{pmatrix} \mathbf{m}_x & \mathbf{m}_y & \mathbf{m}_z \end{pmatrix}^T$ is the model parameter and $\mathbf{m}_x = ln\sigma_x$, $\mathbf{m}_y = ln\sigma_y$, and $\mathbf{m}_z = ln\sigma_z$ are the model parameters in three principal directions; and $\beta_x$, $\beta_y$, and $\beta_z$ are the regularization parameters which are used to balance the data fitting and model constraints.

The data misfit term is described as:

$$\varphi_d(\mathbf{m}) = \| \mathbf{W}_d(\mathbf{d}^{pre} - \mathbf{d}^{obs}) \|^2, \tag{5}$$

where $\mathbf{d}^{pre} = F(\mathbf{m})$ is the predicted data, $F$ is the forward modeling operator, and $\mathbf{d}^{obs}$ represents the observed data. $\mathbf{W}_d$ is the diagonal data weighting matrix, which is constructed based on the reciprocal of the standard error of the observed data, and the readers can refer to Appendix A for more details.

The model regularization term along each principal direction can be written as:

$$\varphi_i = \| \mathbf{L}_i(\mathbf{m}_i - \mathbf{m}_i^{ref}) \|^2, i = x, y, z, \tag{6}$$

where $\mathbf{m}_i^{ref}$ is reference model parameter; and $\mathbf{L}_x$, $\mathbf{L}_y$, and $\mathbf{L}_z$ are the model roughness matrices for the conductivities in the X, Y, and Z directions, respectively. We adopted a method similar to Cai et al., 2021, to calculate the roughness matrix. We can control the smoothness of the inverted model by choosing the number of adjacent elements [29,30]. The roughness matrix can stabilize the inversion by providing a measure of model variations and avoiding spurious structures [31].

The Gauss–Newton method is used to minimize the objective functional, and we solve the normal equation in each Gauss–Newton iteration to obtain the model update:

$$[Re\{(\mathbf{W}_d\mathbf{J})^H\mathbf{W}_d\mathbf{J}\} + \mathbf{L}^T\mathbf{L}]\delta\mathbf{m} \\ = -[Re\{(\mathbf{W}_d\mathbf{J})^H\mathbf{W}_d(\mathbf{d}^{pre} - \mathbf{d}^{obs})\} + \mathbf{L}^T\mathbf{L}(\mathbf{m} - \mathbf{m}^{ref})], \tag{7}$$

$$\mathbf{J} = [\mathbf{J}_x, \mathbf{J}_y, \mathbf{J}_z], \tag{8}$$

$$\mathbf{L} = \begin{bmatrix} \sqrt{\beta_x}\mathbf{L}_x & & \\ & \sqrt{\beta_y}\mathbf{L}_y & \\ & & \sqrt{\beta_z}\mathbf{L}_z \end{bmatrix}, \tag{9}$$

$$\mathbf{m}^{ref} = \left[ \mathbf{m}_x^{ref}, \mathbf{m}_y^{ref}, \mathbf{m}_z^{ref} \right]^T, \tag{10}$$

where $\mathbf{J}_x$, $\mathbf{J}_y$, and $\mathbf{J}_z$ are the sensitivity matrices for anisotropic inversion; $H$ denotes the Hermitian transpose; and $T$ denotes the transpose.

We transform the system of normal equations from the model space to the data space using the Sherman–Morrison–Woodbury formula [32], and the inverse of the reduced Hessian matrix of Equation (7) can be written as:

$$[Re\{(\mathbf{W}_d\mathbf{J})^H\mathbf{W}_d\mathbf{J}\} + \mathbf{L}^T\mathbf{L}]^{-1} = \mathbf{U}^{-1} - \mathbf{U}^{-1}\mathbf{D}\mathbf{\Gamma}^{-1}\mathbf{D}^H\mathbf{U}^{-1}, \tag{11}$$

$$\mathbf{D} = [Re\{(\mathbf{W}_d\mathbf{J})^H\}, Im\{(\mathbf{W}_d\mathbf{J})^H\}], \tag{12}$$

$$\mathbf{U} = \mathbf{L}^T\mathbf{L} + \epsilon \cdot diag\{\mathbf{L}^T\mathbf{L}\}, \tag{13}$$

$$\mathbf{\Gamma} = \mathbf{I} + \mathbf{D}^H\mathbf{U}^{-1}\mathbf{D}. \tag{14}$$

where $\mathbf{I}$ is an identity matrix, and $\epsilon$ is a small positive number, and it is added to Equation (13) to enable the positive definiteness of matrix. Numerical tests show that its influence on the inversion results is negligible [25,26].

We multiply both sides of Equation (7) by the inverse of the reduced Hessian matrix, and substituting Equation (11) into Equation (7) to obtain the system of normal equations in the data space:

$$\delta\mathbf{m} = \mathbf{U}^{-1}\mathbf{R} - \mathbf{U}^{-1}\mathbf{D}\mathbf{\Gamma}^{-1}\mathbf{D}^H\mathbf{U}^{-1}\mathbf{R}, \tag{15}$$

$$\mathbf{R} = -[Re\{(\mathbf{W}_d\mathbf{J})^H\mathbf{W}_d(\mathbf{d}^{pre} - \mathbf{d}^{obs})\} + \mathbf{L}^T\mathbf{L}(\mathbf{m} - \mathbf{m}^{ref})]. \tag{16}$$

For simplicity, Equation (15) is written as:

$$\delta\mathbf{m} = \mathbf{X}_1 - \mathbf{X}_2\mathbf{X}_3, \tag{17}$$

$$\mathbf{X}_1 = \mathbf{U}^{-1}\mathbf{R}, \tag{18}$$

$$\mathbf{X}_2 = \mathbf{U}^{-1}\mathbf{D}, \tag{19}$$

$$\mathbf{X}_3 = \mathbf{\Gamma}^{-1}\mathbf{D}^H\mathbf{X}_1. \tag{20}$$

It is computationally expensive to directly calculate the inverse of matrix $\mathbf{U}$. We obtain $\mathbf{X}_1$ and $\mathbf{X}_2$ by solving the linear equation with $\mathbf{U}$ as the coefficient matrix. $\mathbf{U}$ is a sparse matrix with the size of $N_m \times N_m$ ($N_m$ is the size of model), and it is efficient to obtain $\mathbf{X}_1$ and $\mathbf{X}_2$ using the direct solver. We notice that the size of matrix $\mathbf{D}$ is $2N_d \times N_m$ ($N_d$ is the size of the data set), and we can solve the matrix $\mathbf{X}_2$ by hybrid programming with MPI and OpenMP [33]. The matrix $\mathbf{X}_3$ is solved using the direct solver for dense matrix.

The model-space method requires dealing with the sensitivity matrix $\mathbf{J} \in \mathbb{C}^{N_d \times N_m}$ and the dense matrix of $(\mathbf{W}_d\mathbf{J})^H\mathbf{W}_d\mathbf{J} \in \mathbb{C}^{N_m \times N_m}$. The calculation and storage of the latter term are considerably expensive, and it is generally impractical for the anisotropic inversion. The data-space method deals with matrix $\mathbf{J} \in \mathbb{C}^{N_d \times N_m}$ and $\mathbf{D} \in \mathbb{R}^{2N_d \times N_m}$. Fortunately, these matrices can be stored on the hard disk and some of them can be read into RAM. The matrix of $\mathbf{\Gamma} \in \mathbb{R}^{2N_d \times 2N_d}$ also needs to be stored, but $N_d$ is usually much smaller than $N_m$. Moreover, for the model-space method based on the direct solver, the construction of the coefficient matrix and the right-hand sides of this equation are unsuitable for parallelization. In contrast, most of the solutions of the data space method can be efficiently parallelized.

The regularization parameter is important to obtain geophysical meaningful solutions. We choose the regularization parameters in a similar way as Cai et al., 2021 [30]:

$$\gamma_i = \frac{\parallel Re\{(\mathbf{W}_d\mathbf{J}_i)^H(\mathbf{W}_d\mathbf{J}_i\mathbf{x})\} \parallel_2}{\parallel \mathbf{L}_i^T(\mathbf{L}_i\mathbf{x}) \parallel_2}, i = x, y, z, \tag{21}$$

$$\beta_i = q_i\frac{max\{\gamma_x, \gamma_y, \gamma_z\}}{n_{iter}^c}, i = x, y, z, \tag{22}$$

where $0 < q_i < 1$ is an empirical scaling parameter, $n_{iter}$ is the iteration number, and $c$ is a positive integer that is usually less than 3, based on our numerical tests [34]. We have tested the influence of $q_i$ and $c$ on the inversion results as shown in Appendix B.

Once the model update direction $\delta\mathbf{m}$ is solved using the data-space method [23], a new model is obtained as:

$$\mathbf{m}^{n+1} = \mathbf{m}^n + \alpha\delta\mathbf{m}^n. \tag{23}$$

where $\alpha$ is the step length, which ensures the decreasing of the objective functional. An appropriate step length can be obtained by a simple line search strategy [35].

The definition of RMS in this study is given as follows [36]:

$$\text{RMS} = \sqrt{\frac{(\mathbf{W}_d(\mathbf{d}^{pre} - \mathbf{d}^{obs}))^T(\mathbf{W}_d(\mathbf{d}^{pre} - \mathbf{d}^{obs}))}{N_d}}. \tag{24}$$

The workflow for 3-D triaxial anisotropic inversion in data space is shown in Figure 1.

**Figure 1.** Workflow for 3-D magnetotelluric inversion with triaxial anisotropy.

## 3. Synthetic Model Studies

The developed code is run on the high-performance cluster at China University of Geosciences, Wuhan, China. Each node has 2 Intel Xeon 2.5 GHz CPUs with 20 cores per CPU and 384 GB RAM. We use two cluster nodes in this study. For each node, we set 2 MPI processes and 40 OpenMP threads. The model-space inversion algorithm is solved by the direct solver for dense matrix.

Firstly, we use an example to compare the performance of the model-space and data-space algorithm based on the direct solver. The total number of synthetic observed data is 16,224 (including the real and imaginary parts). We use different discretization to test the performances. The size of models for these different discretization are 20,196, 30,945, 41,280, 61,455, and 82,512, respectively. The memory requirements and CPU time for these models are shown in Figure 2 for the model-space and data-space methods. Although the model-space method is more efficient than the data-space method when the model size is small, the memory consumption and calculation time increase approximately quadratically

with the increase of model size for the model-space method. For the data-space method, the memory consumption and computation time increase linearly with the increase of model size. The model-space method easily becomes impractical because the model parameters can be extremely large. Therefore, the data-space method is more feasible for the large-scale anisotropic inversion in certain scenarios.

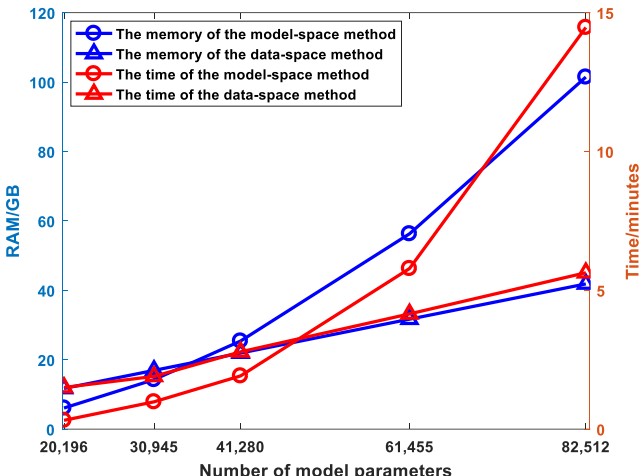

**Figure 2.** The performance of the model-space and the data-space inversion algorithms.

### 3.1. Algorithm Validation

The SM3 model from Cao et al., 2018 [15] is used to demonstrate the correctness of the developed algorithm, as shown in Figure 3. The anisotropic anomaly with the size of 3.6 km × 3.6 km × 1 km is buried in the homogeneous half-space with a resistivity of 300 Ωm. The three principal resistivities of the anisotropic block are $\rho_x/\rho_y/\rho_z = 10\ \Omega\mathrm{m}/30\ \Omega\mathrm{m}/50\ \Omega\mathrm{m}$, and the values of three rotation angles both are 0. The central location of the anomalous body is (0, 0, 1) km. The resistivity of the air layer is chosen as $10^9\ \Omega\mathrm{m}$. We simulate the full impedance tensor data for 5 frequencies (0.1 Hz, 1 Hz, 5 Hz, 10 Hz and 100 Hz) at 169 sites and contaminate the synthetic data by 2% Gauss random noise.

(a)　　　　　　　　　　　　　　　　　　　　　　　　(b)

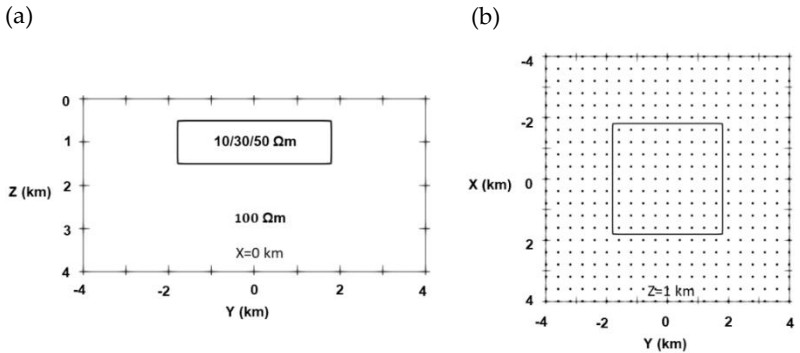

**Figure 3.** Panels (**a**) and (**b**) show the SM3 model from Cao et al., 2018 at *x* = 0 km and *z* = 1 km, respectively. The black dots represent the MT receivers.

The whole simulation domain for the forward modeling and the inversion are the same, and their size are 40 km × 40 km × 40 km in the X, Y, and Z directions. The thickness of the air layer is 20 km for this model. The forward modeling domain is discretized into 743,603 elements. The inversion domain extends from –4 km to 4 km in the horizontal direction and from 0 km to 4 km in the vertical direction. The mesh of the inversion domain is discretized into 171,370 elements, and the total number of elements in the whole mesh is 480,795. Both the initial model and the reference model are set to be a homogeneous half-space with a resistivity of 50 Ωm. The standard error is set to $0.02 \cdot sqrt\left(\left|Z_{xy} \cdot Z_{yx}\right|\right)$. The roughness matrix is calculated using the 20 surrounding elements of each inversion cell.

We set the maximum number of iterations to be 10, and the threshold RMS for the inversion is 1.05.

We terminate the inversion after five iterations when the convergence stalls as shown in Figure 4. The RMS rapidly decrease from the initial value of 16.83 to 1.06. The runtime is 9.2 h and the memory consumption is 137.5 GB.

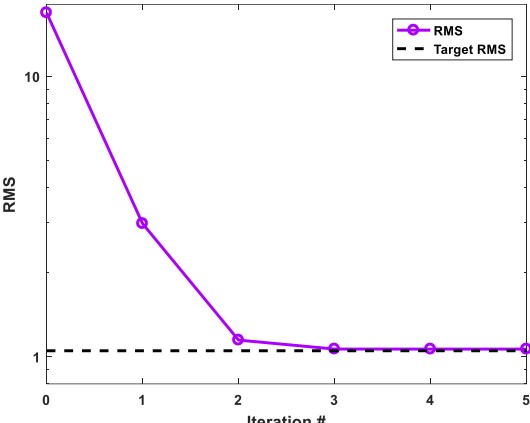

**Figure 4.** Convergence plot for the SM3 model. The black dashed line indicates the threshold RMS.

The anisotropic inversion results are shown in Figures 5–8. The shape and conductivity values in the anisotropic X and Y directions can be recovered well, and the anisotropic coefficient between the resistivity in the X and Y directions (defined by $\log_{10}(\rho_x/\rho_y)$) is also reasonably recovered. The main source of the magnetotelluric field is the plane wave which propagates vertically. Therefore, the resistivity in the Z direction only has limited influence on the MT response. As a result, the resolution of the recovered vertical resistivity is relatively lower compared to that of the horizontal resistivities. [37]. Our inversion results are in a good agreement with Cao et al., 2018 [15].

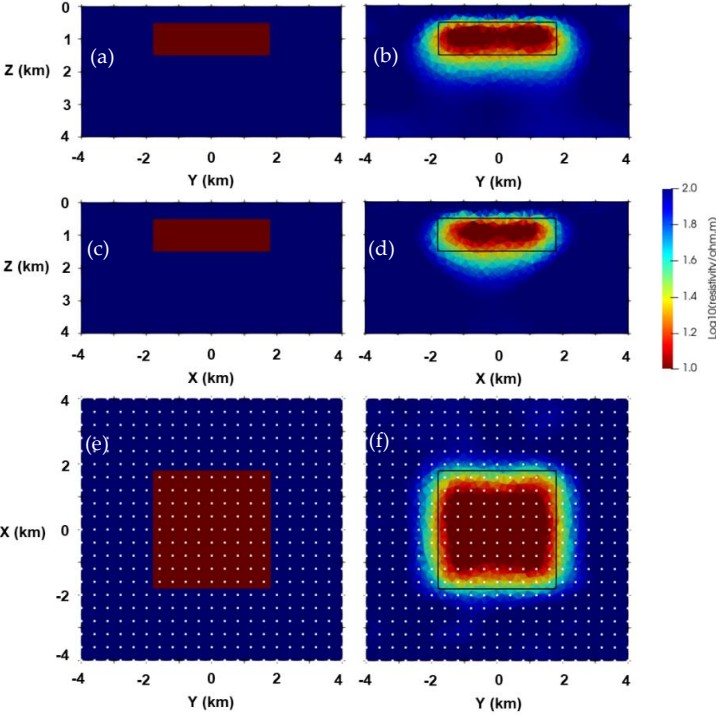

**Figure 5.** The inverted principal conductivity along the X-axis. Panels (**a**) and (**b**) show the true model and the recovered model at $x = 0$ km. Panels (**c**) and (**d**) show the true model and recovered model at $y = 0$ km. Panels (**e**) and (**f**) show the true model and the recovered model at $z = 1$ km.

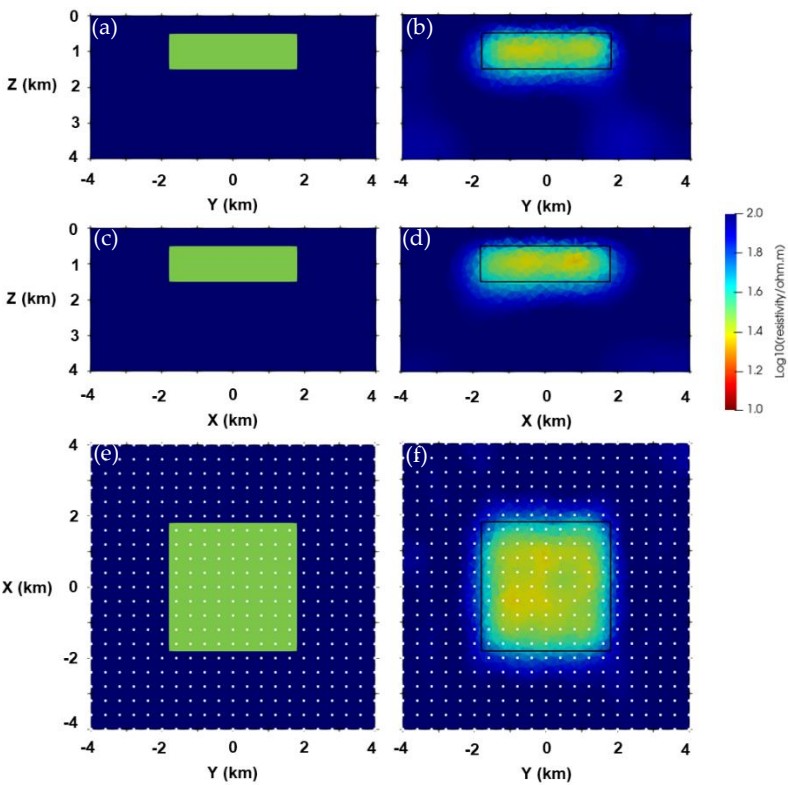

**Figure 6.** The inverted principal conductivity along the Y-axis. Panels (**a**) and (**b**) show the true model and recovered model at $x = 0$ km. Panels (**c**) and (**d**) show the true model and recovered model at $y = 0$ km. Panels (**e**) and (**f**) show the true model and the recovered model at $z = 1$ km.

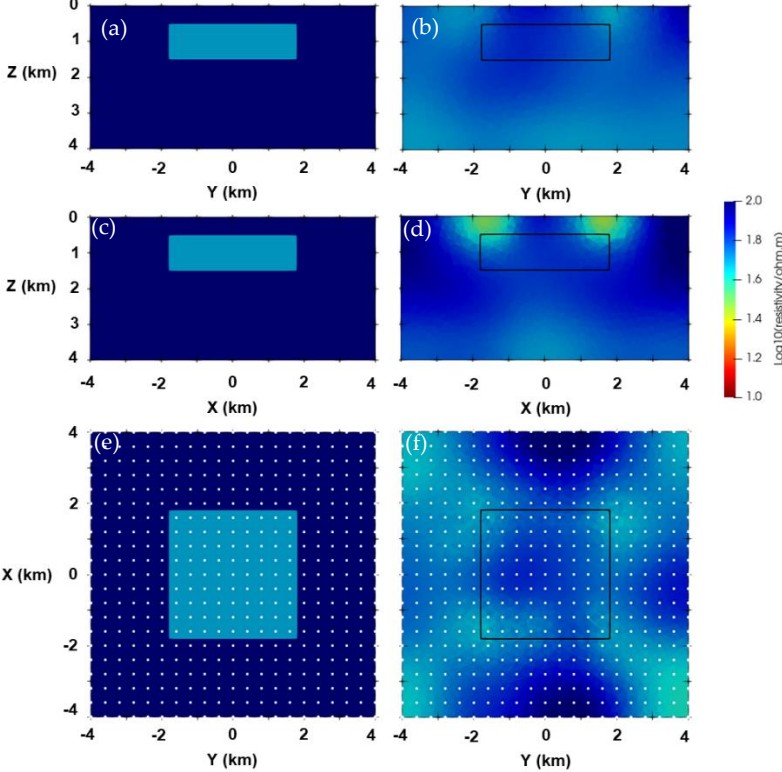

**Figure 7.** The inverted principal conductivity along Z-axis. Panels (**a**) and (**b**) show the true model and the recovered model at $x = 0$ km. Panels (**c**) and (**d**) show the true model and the recovered model at $y = 0$ km. Panels (**e**) and (**f**) show the true model and the recovered model at $z = 1$ km.

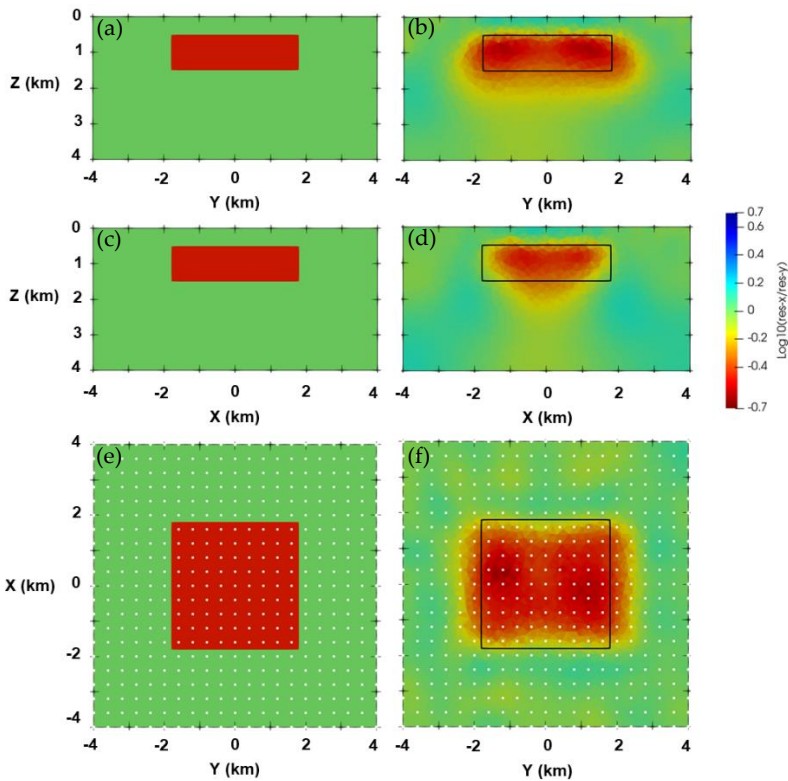

**Figure 8.** The inverted anisotropic coefficient. Panels (**a**) and (**b**) show the true model and the recovered model at $x = 0$ km. Panels (**c**) and (**d**) show the true model and the recovered model at $y = 0$ km. Panels (**e**) and (**f**) show the true model and the recovered model at $z = 1$ km.

The data fitting is shown in Figure 9, we choose the $Z_{xy}$ and $Z_{yx}$ components at the frequency of 5 Hz as an example. We can see that the observed data compare well to the predicted data.

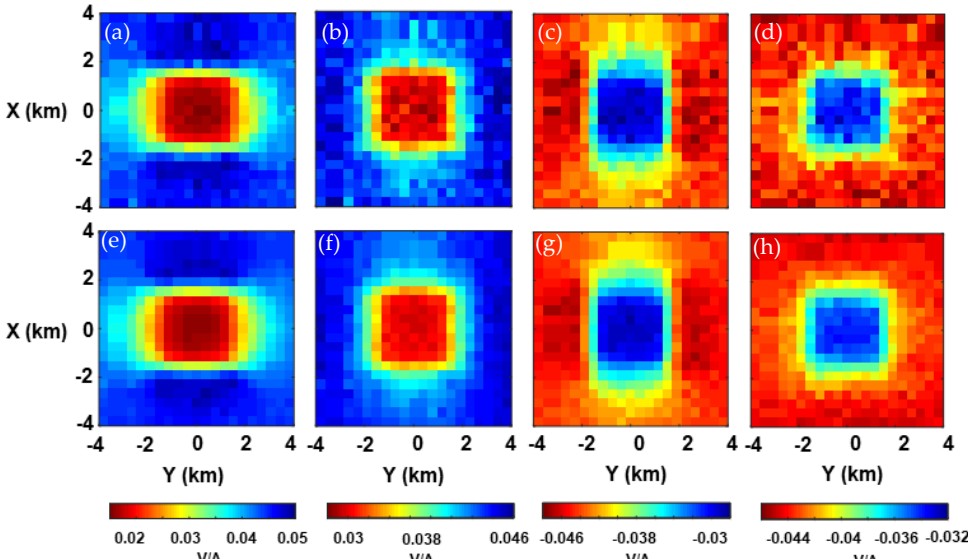

**Figure 9.** Data fitting of $Z_{xy}$ and $Z_{yx}$ components at the frequency of 5 Hz. Panels (**a**) and (**b**) show the real and imaginary parts of the observed $Z_{xy}$ component. Panels (**c**) and (**d**) show the real and imaginary parts of the observed $Z_{yx}$ component. Panels (**e**) and (**f**) show the real and imaginary parts of the predicted $Z_{xy}$ component. Panels (**g**) and (**h**) show the real and imaginary parts of predicted $Z_{yx}$ component.

### 3.2. Two-Blocks Model

To test the effectiveness of our developed inversion algorithm and study the influence of electrical anisotropy, we design a two-blocks model as shown in Figure 10a,b. These two anisotropic anomalies with the resistivities of $\rho_x/\rho_y/\rho_z = 100\ \Omega m/10\ \Omega m/100\ \Omega m$ and $\rho_x/\rho_y/\rho_z = 1000\ \Omega m/10,000\ \Omega m/1000\ \Omega m$ are embedded in the homogeneous half-space with a resistivity of 300 $\Omega m$. Both of these anomalies are 10 km $\times$ 10 km $\times$ 5 km. The central positions of these two bodies are $(-7.5, 0, 4.5)$ km and $(7.5, 0, 4.5)$ km, respectively. The full impedance tensor data for 12 frequencies, evenly spaced in the logarithmic space from 0.001 Hz to 10 Hz, at 169 sites are used as the observed data. We contaminate the synthetic data with 2% Gauss random noise. In Appendix C, we also study the influence of different noise levels on the inversion results.

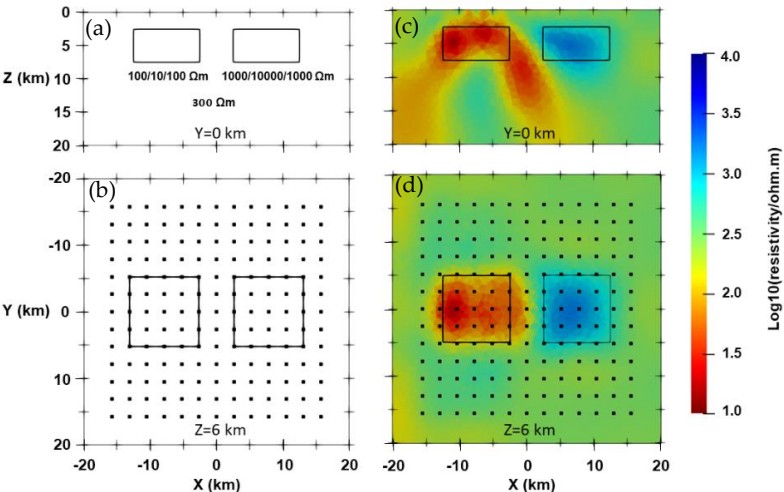

**Figure 10.** Panels (**a**) and (**b**) show the true model at $y = 0$ km and $z = 6$ km. Panels (**c**) and (**d**) show the isotropic inversion results at $y = 0$ km and $z = 6$ km. The black dots represent MT receivers.

The size of the whole simulation domain for the forward modeling and inversion are both 200 km $\times$ 200 km $\times$ 100 km. The domain extends from $-100$ km to 100 km in the horizontal direction and from $-20$ km to 80 km in the vertical direction. The mesh for forward modeling is discretized into 834,627 cells. For the inversion, the mesh in the inversion domain is discretized into 188,389 cells, and the total number of mesh is 417,504. The inversion domain extends from $-20$ km to 20 km in the horizontal direction and from 0 km to 20 km in the vertical direction. The initial and reference models are set to be 100 $\Omega m$ homogeneous half-space. The standard error is set to be $0.02 \cdot sqrt(|Z_{xy} \cdot Z_{yx}|)$. The roughness matrix is calculated by the 20 surrounding elements of each cell. The maximum number of inversion iterations is set to 10, and the threshold RMS for the inversion is 1.05.

The model difference between the true model and the inverted model is defined as follows [38]:

$$\delta = \sqrt{\frac{\sum_{i=1}^{N_m}\left(m_i^{inv} - m_i^{true}\right)^2}{N_m}},$$  (25)

where $m_i^{inv}$ and $m_i^{true}$ are the inverted model and the true model.

The isotropic inversion results are shown in Figure 10c,d. From this figure, we can see that the isotropic inversion produces some artifacts surrounding the anisotropic conductive body. Moreover, the shape and location of the conductive body are poorly recovered. The shape and location of the anisotropic resistive body can be recovered in this scenario, which reveals that the effect of electrical anisotropic on the resistive body is smaller than the conductive body [36]. The anisotropic inversion results are shown in Figures 11 and 12. From these two figures, we can see that the shapes and locations of these two anisotropic anomalies are well recovered in the principal X and Y directions. The anisotropic coefficient between the resistivity in the principal X and Y directions is also reasonably recovered.

Considering the resistivity in the Z direction cannot be effectively recovered in anisotropic inversion, we calculate the model difference $\delta$ using the resistivity in the X and Y direction. The model difference $\delta$ is 0.94 and 0.75 for isotropic and anisotropic inversion, respectively. With anisotropic inversion, we can obtain a better data fitting for this model.

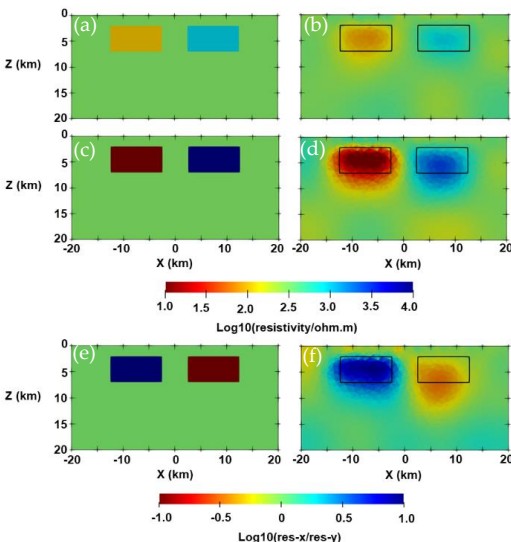

**Figure 11.** Anisotropic inversion results at $y = 0$ km. Panels (**a**), (**c**), and (**e**) show the true model for the principal X, Y directions and the anisotropic coefficient. Panels (**b**), (**d**), and (**f**) show the corresponding inversion results.

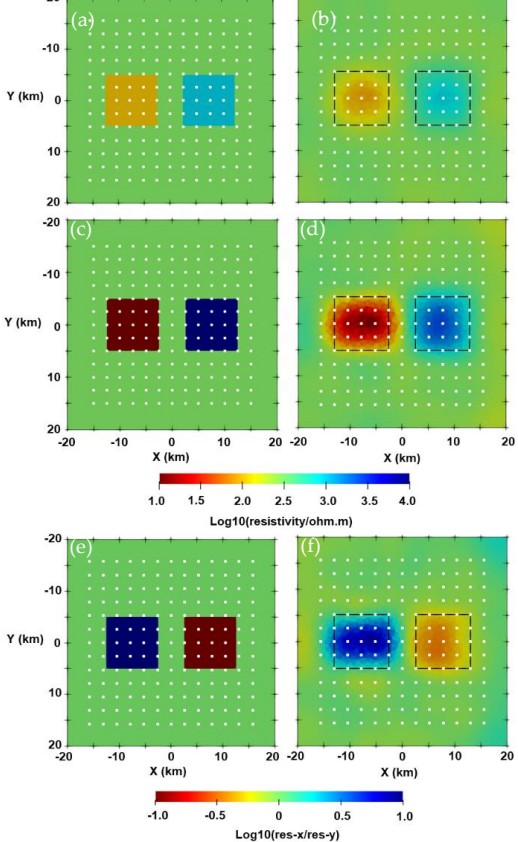

**Figure 12.** Anisotropic inversion results at $z = 6$ km. Panels (**a**), (**c**), and (**e**) show the true model for the principal X and Y directions, and the anisotropic coefficient. Panels (**b**), (**d**), and (**f**) show the corresponding inversion results.

The convergence plots for the isotropic and anisotropic inversion are shown in Figure 13. We terminate both inversions after five iterations when the convergence stalls. For the isotropic inversion, the RMS decrease from the initial value of 23.57 to 1.97. The runtime is 5.6 h and the memory consumption is 45.6 GB. For the anisotropic inversion, the RMS decrease from the initial value of 23.57 to 1.12. The runtime is 9.7 h and the memory consumption is 138.7 GB. If the model space method was used for the anisotropic inversion, the memory requirement would be 4759.6 GB.

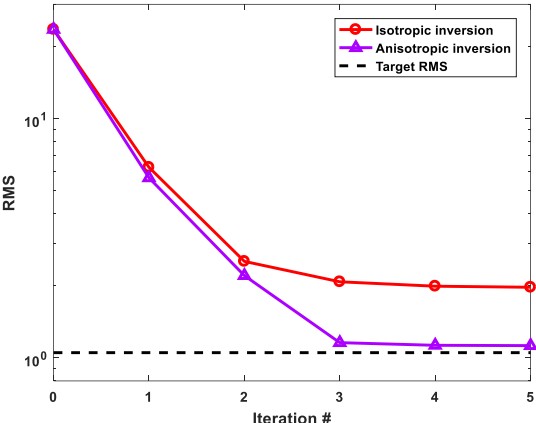

**Figure 13.** Convergence plot for the isotropic and anisotropic inversion. The black dashed line indicates the threshold RMS.

We show the data fitting for isotropic and anisotropic inversion in Figure 14. The $Z_{xy}$ and $Z_{yx}$ components at the frequency of 0.46 Hz are chosen as an example. It is clear that the data fitting for the anisotropic inversion is better than that of the isotropic inversion.

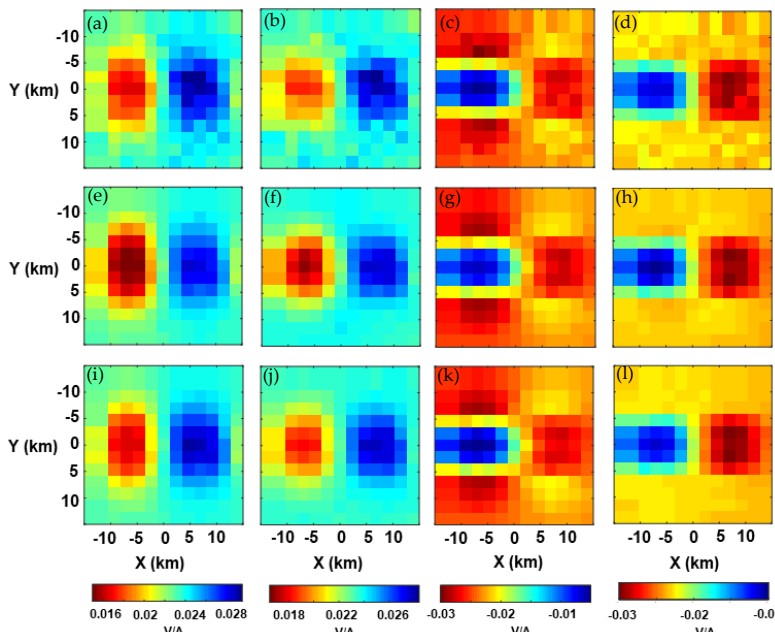

**Figure 14.** Data fitting for $Z_{xy}$ and $Z_{yx}$ components at the frequency of 0.46 Hz. Panels (**a**) and (**b**) show the real and imaginary parts of observed $Z_{xy}$ component. Panels (**c**) and (**d**) show the real and imaginary parts of observed $Z_{yx}$ component. Panels (**e**) and (**f**) show the real and imaginary parts of predicted $Z_{xy}$ component for isotropic inversion. Panels (**g**) and (**h**) show the real and imaginary parts of predicted $Z_{yx}$ component for isotropic inversion. Panels (**i**) and (**j**) show the real and imaginary parts of predicted $Z_{xy}$ component for anisotropic inversion. Panels (**k**) and (**l**) show the real and imaginary parts of predicted $Z_{yx}$ component for anisotropic inversion.

We know that MT only has limited sensitivity to the vertical resistivity. As a result, we may exclude the vertical resistivity from the model parameters. We examine the effect of including/excluding the vertical resistivity in the inversion parameters, as shown in Figures 15 and 16. We can see that the inversion results are almost the same when including and excluding the vertical resistivity from the model parameters. When excluding the vertical resistivity from the inversion model parameters, we assume its value is the same as the initial horizontal resistivity. The model difference $\delta$ are 0.75 and 0.74 for the inversion with including/excluding the vertical resistivity from the model parameters. Therefore, we do not consider the influence of vertical resistivity in the later section.

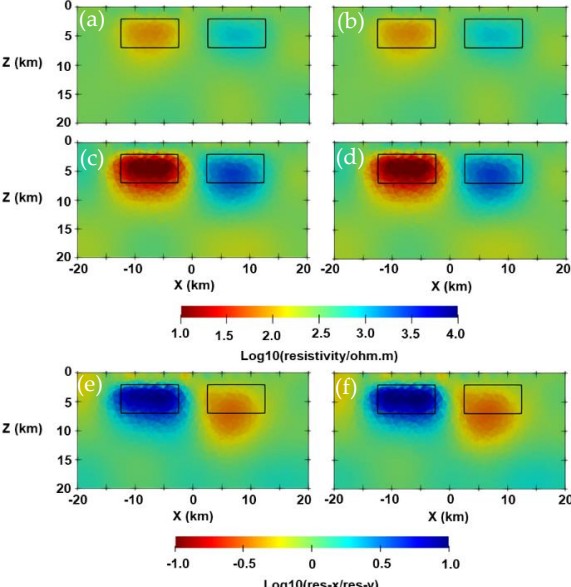

**Figure 15.** Anisotropic inversion results at $y = 0$ km. Panels (**a**) and (**b**) show the inverted resistivity in the principal X direction when including and excluding the vertical resistivity in the inversion. Panels (**c**) and (**d**) show the inverted resistivity in the principal Y direction when including and excluding the vertical resistivity in the inversion. Panels (**e**) and (**f**) show the inverted anisotropic coefficient when including and excluding the vertical resistivity in the inversion.

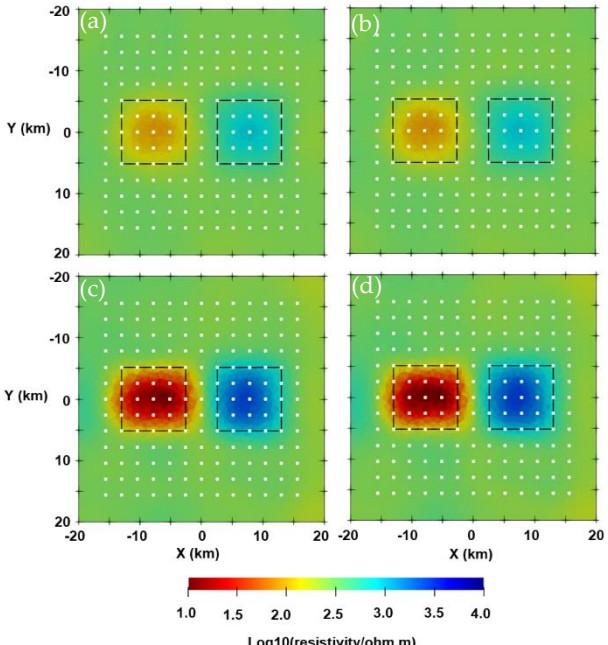

**Figure 16.** *Cont.*

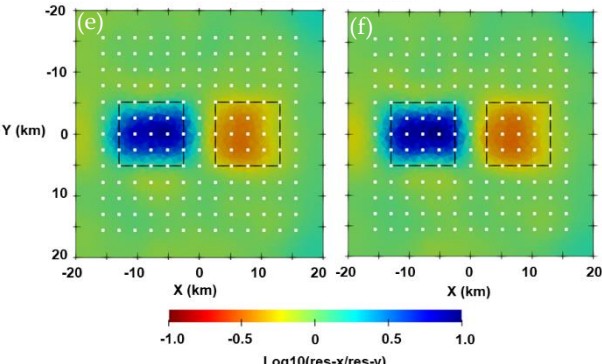

**Figure 16.** Anisotropic inversion results at *z* = 6 km. Panels (**a**) and (**b**) show the inverted resistivity in the principal X direction when including and excluding the vertical resistivity in the inversion. Panels (**c**) and (**d**) show the inverted resistivity in the principal Y direction when including and excluding the vertical resistivity in the inversion. Panels (**e**) and (**f**) show the inverted anisotropic coefficient when including and excluding the vertical resistivity in the inversion.

### 3.3. Isotropic and Anisotropic Blocks Model with Topography

To further test the performance of the developed algorithm, we design an isotropic and anisotropic blocks model, as shown in Figure 17. The background model is a homogeneous half-space with the resistivity of 300 $\Omega$m. The three principal resistivities of the anisotropic block are $\rho_x/\rho_y/\rho_z = 10\ \Omega$m$/100\ \Omega$m$/10\ \Omega$m, and its central location is (0, −7.5, 5.5) km. The resistivity of the isotropic block is 1000 $\Omega$m, and its central location is (0, 7.5, 5.5) km. The full impedance tensor data for 12 frequencies, evenly spaced in the logarithmic scale from 0.01 Hz to 100 Hz, at 169 sites are used for inversion. We add 2% Gauss random noise to the synthetic data. The topography in the inversion domain is shown in Figure 18.

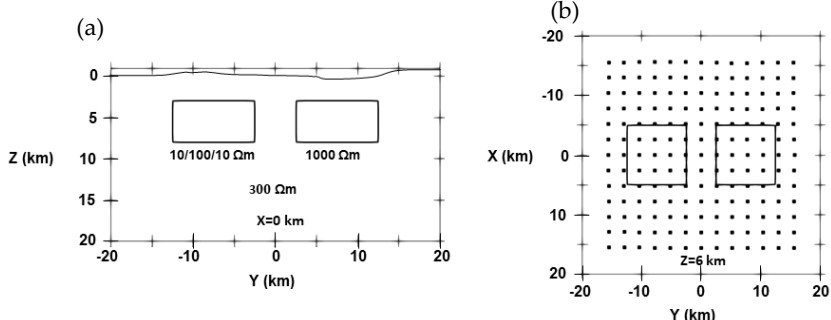

**Figure 17.** Panels (**a**) and (**b**) show the isotropic and anisotropic blocks model at *x* = 0 km and *z* = 6 km, respectively. The black dots represent the MT receivers.

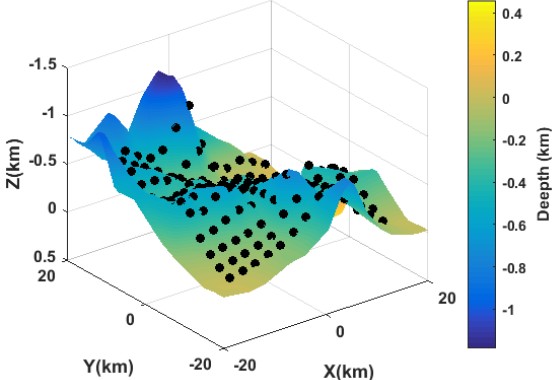

**Figure 18.** The topography in the inversion domain (with exaggeration in the vertical direction). The black dots indicate the MT receivers.

The whole simulation domain extends from $-100$ km to $100$ km in the horizontal direction and from $-20$ km to $80$ km in the vertical direction. The inversion domain extends from $-20$ km to $20$ km in the horizontal direction and from $-1.16$ km to $20$ km in the vertical direction. The mesh for forward modeling is discretized into 824,132 cells. The inversion domain is discretized into 200,934 cells, and the total number of cells for the inversion mesh is 644,824. The initial and reference models of the inversion are a homogeneous half-space with the resistivity of 100 $\Omega$m. The standard error is set to $0.02 \cdot sqrt\left(\left|Z_{xy} \cdot Z_{yx}\right|\right)$. The maximum number of inversion iterations is set to 10, and the threshold RMS for the inversion is 1.05.

The convergence plot is shown in Figure 19, we can see that the RMS decreases from 21.72 to 1.05 after five iterations, the runtime is 8.2 h and the memory consumption is 99.2 GB.

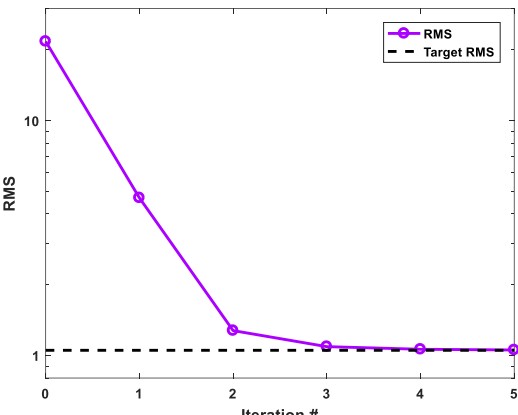

**Figure 19.** Convergence plot for the isotropic and anisotropic blocks model with topography. The black dashed line indicates the threshold RMS.

The inversion results are shown in Figures 20 and 21, we can see that the shape and position of the isotropic and anisotropic blocks are well recovered. The inversion does not produce obvious artifacts for the isotropic anomalous body.

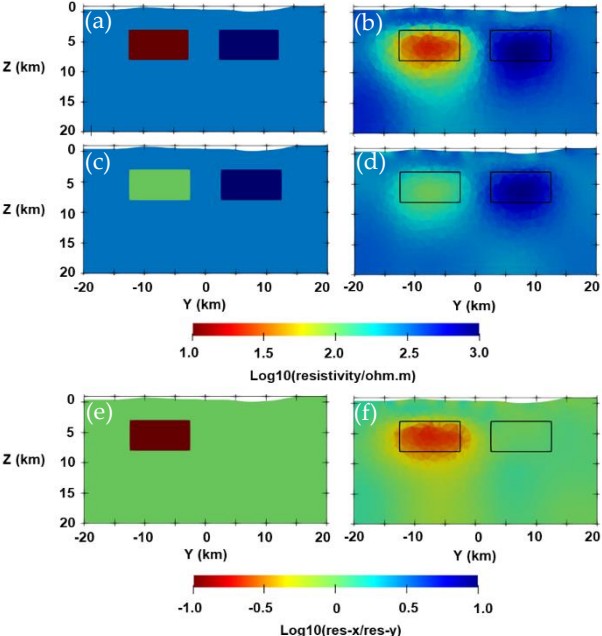

**Figure 20.** Anisotropic inversion results at $x = 0$ km. Panels (**a**), (**c**), and (**e**) show the true model for principal X and Y directions, and the anisotropic coefficient. Panels (**b**), (**d**), and (**f**) show the anisotropic inversion results for the principal X and Y directions, and the anisotropic coefficient.

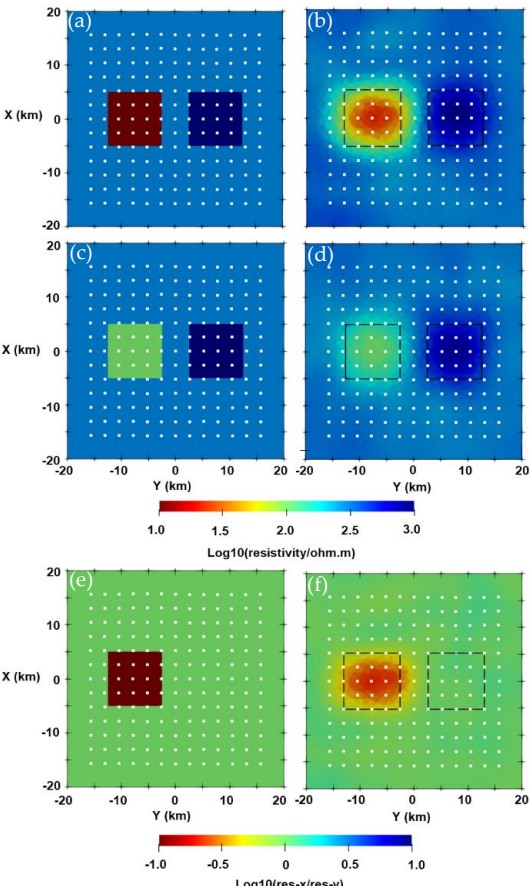

**Figure 21.** Anisotropic inversion results at *z* = 5.5 km. Panels (**a**), (**c**), and (**e**) show the true model for the principal X and Y directions, and the anisotropic coefficient. Panels (**b**), (**d**), and (**f**) show the anisotropic inversion results for the principal X and Y directions, and the anisotropic coefficient.

The data fitting is shown in Figure 22, we choose the $Z_{xy}$ and $Z_{yx}$ components at the frequency of 0.285 Hz as an example. We can see that the observed data compare well with the predicted data.

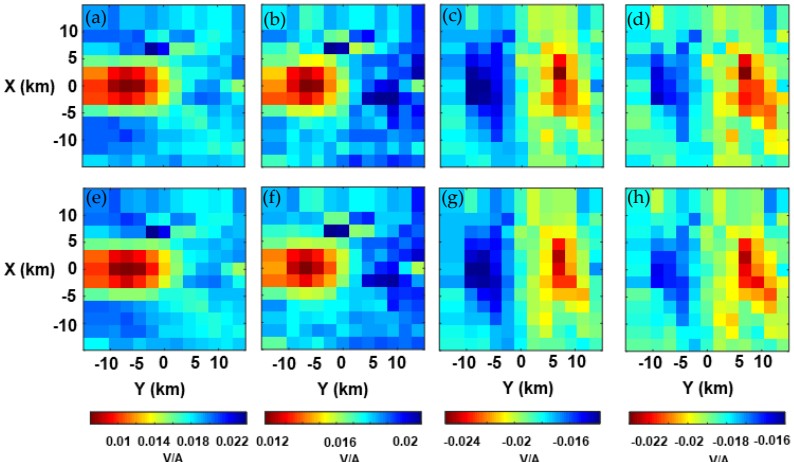

**Figure 22.** Data fitting for $Z_{xy}$ and $Z_{yx}$ components at the frequency of 0.285 Hz. Panels (**a**) and (**b**) show the real and imaginary parts of observed $Z_{xy}$ component. Panels (**c**) and (**d**) show the real and imaginary parts of the observed $Z_{yx}$ component. Panels (**e**) and (**f**) show the real and imaginary parts of the predicted $Z_{xy}$ component. Panels (**g**) and (**h**) show the real and imaginary parts of the predicted $Z_{yx}$ component.

## 4. Conclusions

In this paper, we develop a triaxial anisotropic 3-D MT inversion algorithm with unstructured tetrahedral mesh. To reduce the memory requirement, we transform the algorithm from the model space to the data space and derived the inversion workflow using the Gauss–Newton approach. In the data-space inversion, the model update direction can be effectively solved by the parallel direct solver.

We compare the efficiency of the model-space and data-space methods by an example. We found that the data-space algorithm requires much less memory and computation time than the model space method when the number of model parameters is large. We demonstrate the effectiveness and applicability of the anisotropic inversion algorithm by several synthetic models. The electrical structures in the principal X and Y directions can be well recovered, but the resistivity of the principal Z direction cannot be reasonably recovered by the current inversion algorithm. We will apply the developed anisotropic inversion algorithm in data space to the field data in the future research.

**Author Contributions:** Conceptualization, J.X., H.C. and X.H.; methodology, J.X. and H.C.; software, H.C. and J.X.; validation, S.H. and M.L.; formal analysis, H.C. and J.X.; resources, X.H. and H.C.; writing—original draft preparation, J.X.; writing—review and editing, H.C.; supervision, X.H. and H.C. All authors have read and agreed to the published version of the manuscript.

**Funding:** This research was funded by National Natural Science Foundation of China, grant number 41974089.

**Data Availability Statement:** Not applicable.

**Acknowledgments:** We acknowledge the support of this work from the computation center at China University of Geosciences (Wuhan).

**Conflicts of Interest:** The authors declare no conflict of interest.

## Appendix A

The data weighted matrix $\mathbf{W}_d$ usually is used with nonuniform and uniform data variance [39], and other forms. We tested the effect of $\mathbf{W}_d$ with nonuniform and uniform data variance using the synthetic data (with 2% Gauss random noise) using the two-blocks model in Section 3.2.

The data weighted matrix $\mathbf{W}_d$ can be written as follows:

$$\mathbf{W}_d = \begin{bmatrix} 1/e_1 & & & \\ & 1/e_2 & & \\ & & \ddots & \\ & & & 1/e_{N_d} \end{bmatrix}, \tag{A1}$$

where $e_i$ is the data variance of the ith observed data.

The nonuniform data variance can be written as follows [18,20,24]:

$$e(Z_{ij}) = \varepsilon \cdot \sqrt{|Z_{xy} \cdot Z_{yx}|}, \; i,j = x,y. \tag{A2}$$

where $\varepsilon$ is error floor, we choose $\varepsilon = 0.02$ in this case.

The uniform data variance can be written as [24]:

$$e(Z_{ij}) = max\{3.5\% \cdot (|Z_{xy} - Z_{yx}|/2)\}, \; i,j = x,y. \tag{A3}$$

These inversion results are shown in Figures A1 and A2, and we reset the color bar for a higher contrast. The RMS for $\mathbf{W}_d$ with nonuniform and uniform data variance are 1.12 and 1.17, and the model difference $\delta$ are 0.75 and 0.76, respectively.

We can see that these two anisotropic anomalies are both well recovered for $\mathbf{W}_d$ with nonuniform and uniform data variance. However, the conductive artifact blows the

resistive body, for $\mathbf{W}_d$ with uniform data variance, more seriously than that for $\mathbf{W}_d$ with nonuniform data variance. Moreover, the anisotropic coefficient for $\mathbf{W}_d$ with nonuniform data variance is better recovered.

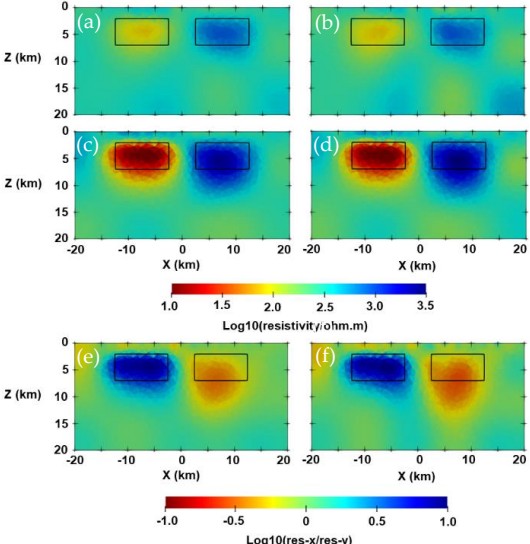

**Figure A1.** Anisotropic inversion results at $y = 0$ km. Panels (**a**) and (**b**) show the inverted resistivity in the principal X direction for $\mathbf{W}_d$ with nonuniform and uniform data variance. Panels (**c**) and (**d**) show the inverted resistivity in the principal Y direction for $\mathbf{W}_d$ with nonuniform and uniform data variance. Panels (**e**) and (**f**) show the inverted anisotropic coefficient for $\mathbf{W}_d$ with nonuniform and uniform data variance.

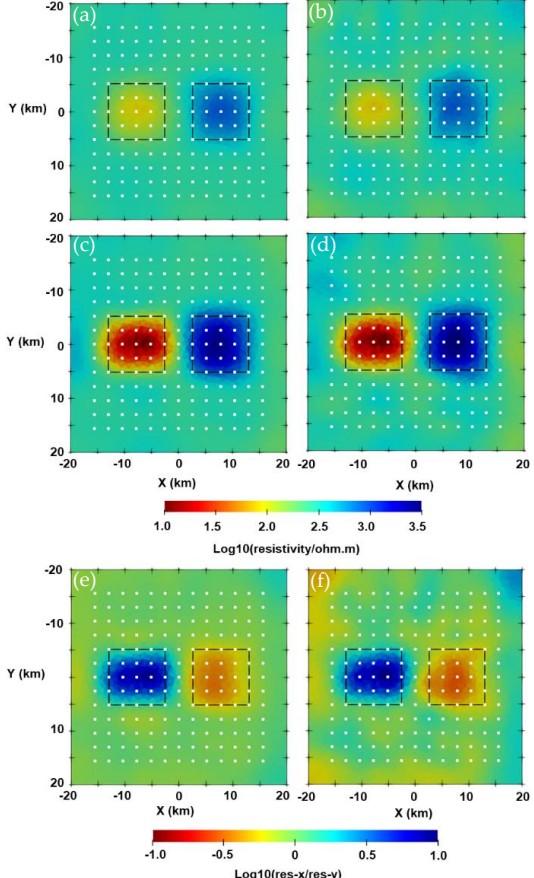

**Figure A2.** Anisotropic inversion results at $z = 6$ km. Panels (**a**) and (**b**) show the inverted resistivity

in the principal X direction for $\mathbf{W}_d$ with nonuniform and uniform data variance. Panels (**c**) and (**d**) show the inverted resistivity in the principal Y direction for $\mathbf{W}_d$ with nonuniform and uniform data variance. Panels (**e**) and (**f**) show the inverted anisotropic coefficient for $\mathbf{W}_d$ with nonuniform and uniform data variance.

### Appendix B

We also carried out some numerical tests on the choices of $q_i$ and $c$ using the synthetic data (with 2% Gauss random noise) of the two-blocks model in Section 3.2, and the standard error is set to $0.02 \cdot sqrt(|Z_{xy} \cdot Z_{yx}|)$. The final RMS for the case with $q_i = 0.05$, 0.1, 0.4, 0.8 are 1.05, 1.06, 1.09, 1.15, and the model difference $\delta$ are 0.78, 0.75, 0.75, 0.76. The final RMS for $c = 1$, 2, 3 are 1.12, 1.06, 1.04, and the model difference $\delta$ are 0.75, 0.75, 0.79.

We can see from Figures A3 and A4 that the recovered background model is inaccurate when $q_i = 0.05$, but these two anisotropic bodies can be reasonably recovered. Although these inversion results are similar, the background model can be better recovered when $q_i$ increases from 0.1 to 0.8. Taking the above factors into consideration, we believe that $q_i = 0.8$ is appropriate.

We can see from Figures A5 and A6 that these inversion results are similar, but the larger value of $c$, the worse the inverted background model. As a result, we choose $c = 1$ in this case.

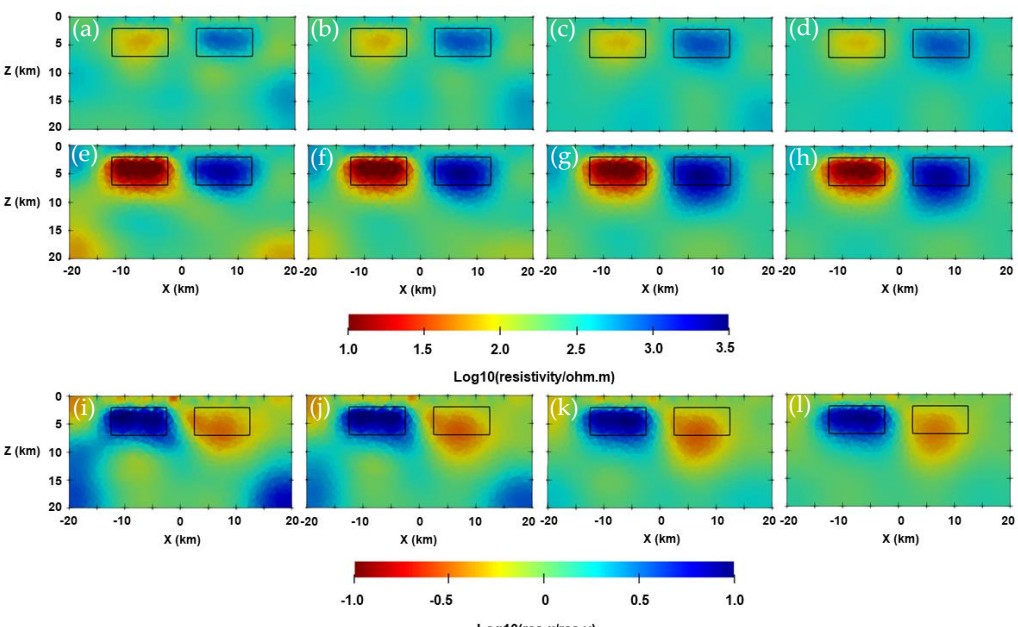

**Figure A3.** Anisotropic inversion results at $y = 0$ km for different $q_i$, and $c = 1$. Panels (**a**)–(**d**) show the inverted resistivity in the principal X direction for $q_i = 0.05$, 0.1, 0.4, 0.8. Panels (**e**)–(**h**) show the inverted resistivity in principal Y direction for $q_i = 0.05$, 0.1, 0.4, 0.8. Panels (**i**)–(**l**) show the inverted anisotropic coefficient for $q_i = 0.05$, 0.1, 0.4, 0.8.

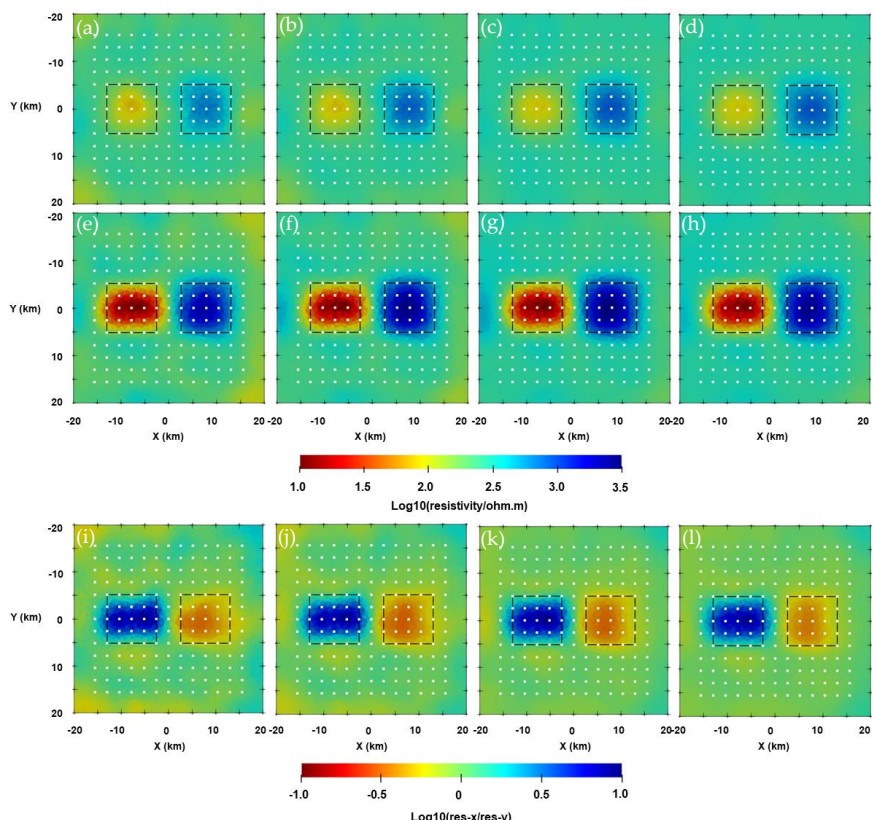

**Figure A4.** Anisotropic inversion results at $z = 6$ km for different $q_i$, and $c = 1$. Panels (**a**)–(**d**) show the inverted resistivity in the principal X direction for $q_i = 0.05$, $0.1$, $0.4$, $0.8$. Panels (**e**)–(**h**) show the inverted resistivity in principal Y direction for $q_i = 0.05$, $0.1$, $0.4$, $0.8$. Panels (**i**)–(**l**) show the inverted anisotropic coefficient for $q_i = 0.05$, $0.1$, $0.4$, $0.8$.

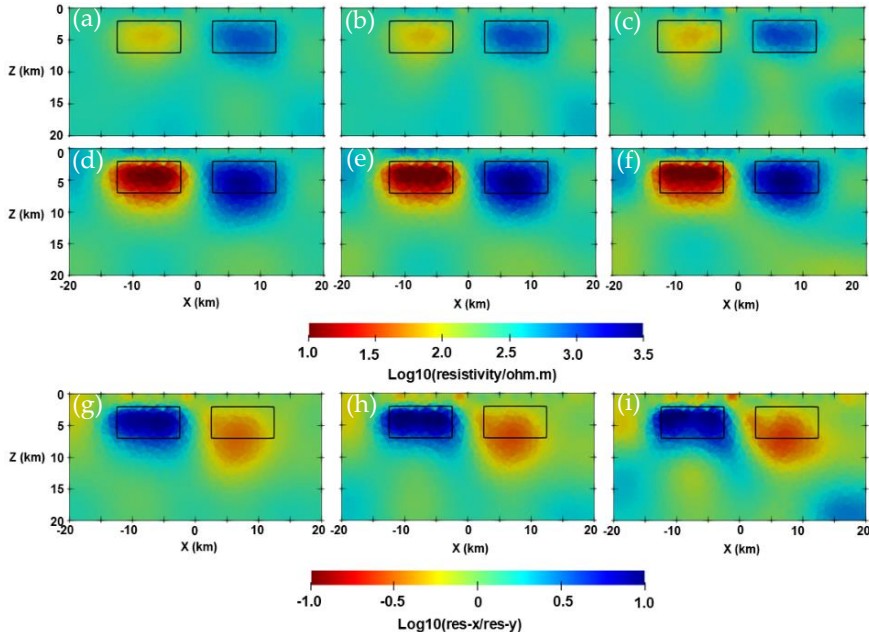

**Figure A5.** Anisotropic inversion results at $y = 0$ km for different $c$, and $q_i = 0.6$. Panels (**a**)–(**c**) show the inverted resistivity in the principal X direction for $c = 1, 2, 3$. Panels (**d**)–(**f**) show the inverted resistivity in the principal Y direction for $c = 1, 2, 3$. Panels (**g**)–(**i**) show the inverted anisotropic coefficient for $c = 1, 2, 3$.

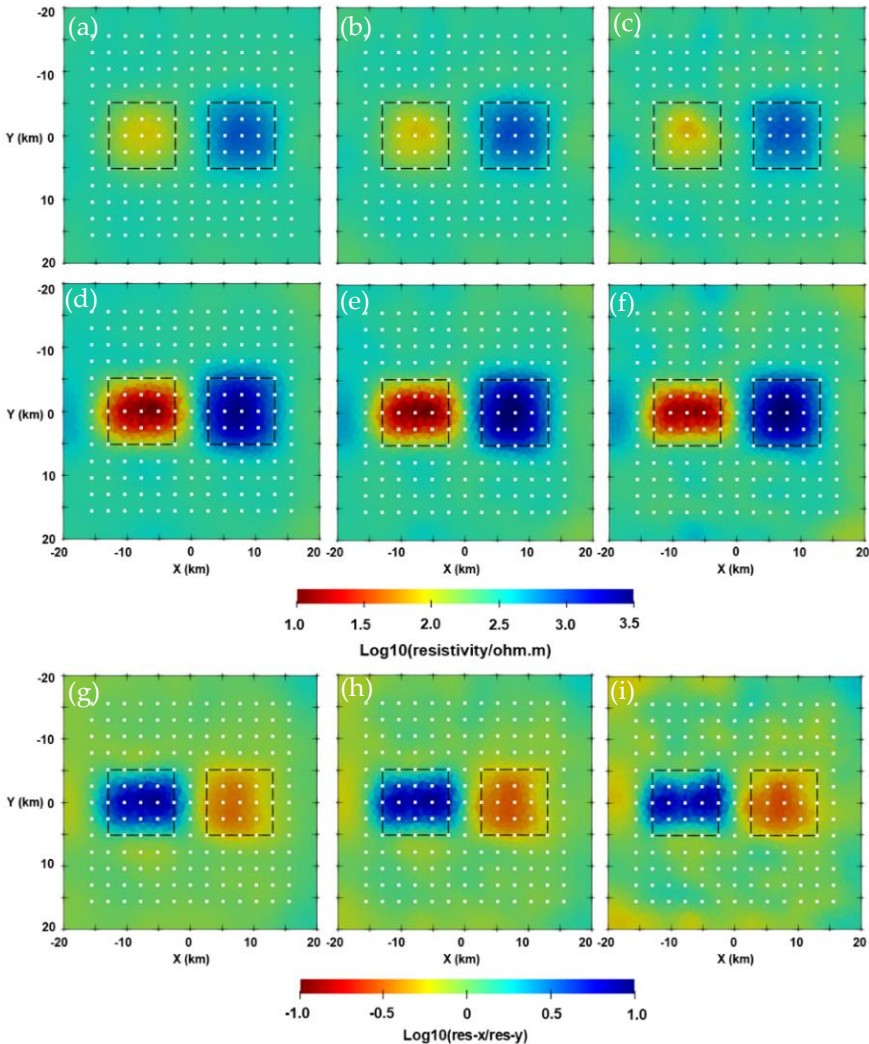

**Figure A6.** Anisotropic inversion results at $z = 6$ km for different $c$, and $q_i = 0.6$. Panels (**a**)–(**c**) show the inverted resistivity in the principal X direction for $c = 1, 2, 3$. Panels (**d**)–(**f**) show the inverted resistivity in the principal Y direction for $c = 1, 2, 3$. Panels (**g**)–(**i**) show the inverted anisotropic coefficient for $c = 1, 2, 3$.

## Appendix C

We test the effects of noise level on the inversion results in this section. In this study, 2%, 5%, and 8% Gauss random noise are added to synthetic observed data for the two-blocks model in Section 3.2.

The inversion results are shown in Figures A7 and A8. We can see that although the shape and location of these two anisotropic anomalies are well recovered for the resistivity in the principal X and Y directions, and the anisotropic coefficient, the recovered background model is distorted when the noise increases. The convergence stalls after five iterations for all cases. The RMS are 1.12, 2.65, and 4.24, and the model difference $\delta$ are 0.75, 0.76, and 0.77 when 2%, 5%, and 8% random noise are added to observed data.

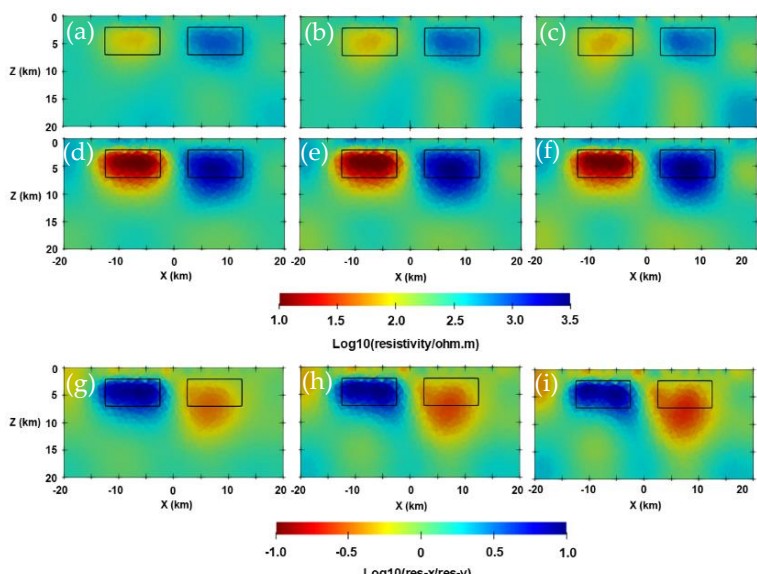

**Figure A7.** Anisotropic inversion results at *y* = 0 km. Panels (**a**)–(**c**) show the inverted resistivity in the principal X direction for 2%, 5%, 8% noise level. Panels (**d**)–(**f**) show the inverted resistivity in the principal Y direction for 2%, 5%, 8% noise level. Panels (**g**)–(**i**) show the inverted anisotropic coefficient for 2%, 5%, 8% noise level.

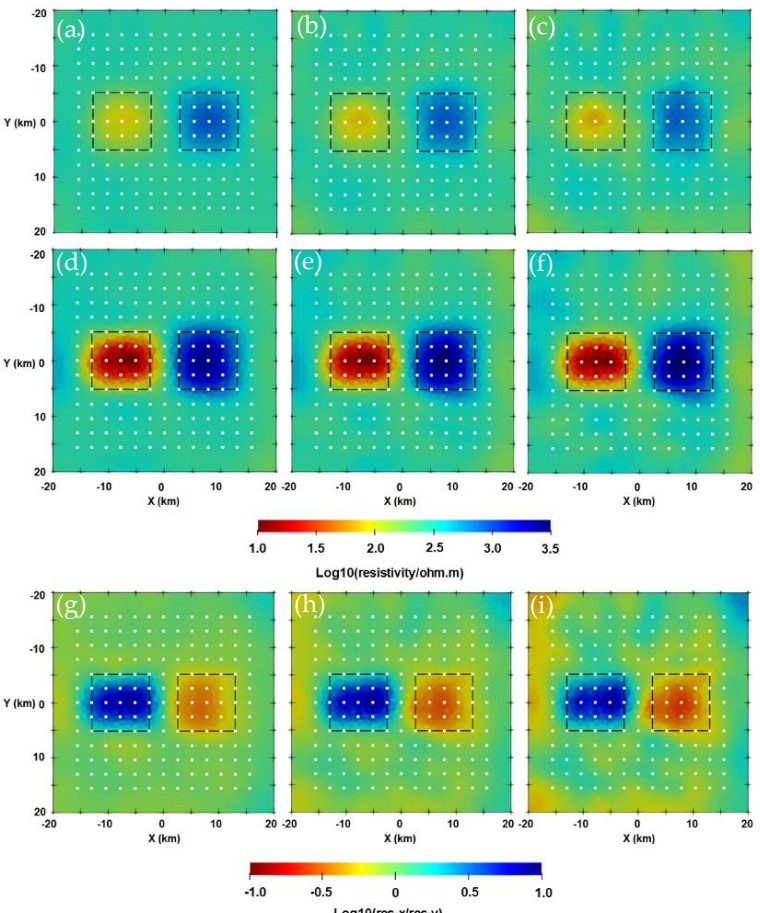

**Figure A8.** Anisotropic inversion results at *z* = 6 km. Panels (**a**)–(**c**) show the inverted resistivity in the principal X direction for 2%, 5%, 8% noise level. Panels (**d**)–(**f**) show the inverted resistivity in the principal Y direction for 2%, 5%, 8% noise level. Panels (**g**)–(**i**) show the inverted anisotropic coefficient for 2%, 5%, 8% noise level.

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
