# Peer review of "Three-Dimensional Magnetotelluric Inversion for Triaxial Anisotropic Medium in Data Space"

_minerals, doi:10.3390/min12060734_

Round 1

Reviewer 1 Report

The paper is generally written well. However, there are a few points that should be corrected as follows:

There is no need to put a dot between omega and m for resistivity values. 

Line 48-49: No verb in the sentences. It may be read: ‘The stability of iterative solvers decreases because the 3-D anisotropic magnetotelluric inversion is an ill-conditioned problem.

Line 209: ‘finish misfit’ should read ‘final misfit’. The sentences should be corrected as  ‘the final misfit is 0.002 after four iterations and …’

Author Response

(1) There is no need to put a dot between omega and m for resistivity values.

Reply: The issue has been addressed in the revision.

(2) Line 48-49: No verb in the sentences. It may be read: ‘The stability of iterative solvers decreases because the 3-D anisotropic magnetotelluric inversion is an ill-conditioned problem.’

Reply: We have corrected this problem, please refer to page 2, line 49-51 in the revised manuscript for this revision.

(3) Line 209: ‘finish misfit’ should read ‘final misfit’. The sentences should be corrected as ‘the final misfit is 0.002 after four iterations and … ’

Reply: We have corrected this problem, please refer to page 8, line 212-214 in the revised manuscript for this revision.

Reviewer 2 Report

Comments on “Three-dimensional magnetotelluric inversion for triaxial anisotropic medium in data space” by Xie et al.

Recommendation

This paper deals with full 3-D anisotropic magnetotelluric inversion. Such method will be useful to reveal anisotropic property in the Earth, and therefore the topic is important and interesting for general readers of Minerals. However, I can find a lot of shortcomings in the present manuscript which is neither well organized nor well written. I would recommend that this paper has to be evaluated again after completely re-organized and rewritten by taking following comments into account. The authors say that introducing data space inversion is a new contribution by the present paper, but I was not convinced as shown in my comments below. Data space inversion is well known technology, and it is doubtful whether it brings essential advantage to the inversion problem of present concern. I suggest the authors to add new contents that can be regarded as a unique and significant contribution to this problem. If the authors fail to add new contents, the present paper may not be acceptable. Also editing English by a native speaker is indispensable.

Major comments

  • A very similar study (Kong et al., 2022) has been published. Although the former work is cited in this paper (line 44-47), only a difference between these two studies is pointed out briefly. After reading the cited paper, I realized that the present paper is mostly repeating the work of Kong et al. (2022). The basic theory is essentially the same. The description of inversion problem is nearly the same except technical aspects (e.g., inversion process is converted from model to data space in the present paper). Synthetic study is also conducted in a similar manner, and similar results are presented. It is strange that synthetic models are just similar but not exactly the same, and therefore it is impossible to quantitatively compare the performance. The authors must try to cite past studies wherever applicable and clarify where the present study is different from them, so that they can clearly show how the present study makes unique contribution as pointed out below. However, synthetic models should not be slightly different from Kong et al. (2022). They should be exactly the same and the inversion performance should be compered.
  • As for the basic theory, both this paper and Kong et al. paper derive basic equations of EM induction in a general anisotropic medium, in which the anisotropic conductivity is expressed by a real valued 3x3 tensor with degree of freedom 6 (3 principle conductivity values and 3 Euler angles). However, the basic theory of Kong et al. (2022) apparently treats Euler angle as independent parameters but implicitly assumed to be fixed in practice. Although the same assumption is made and explicitly stated in this paper (line 82), it is questionable whether the principle axes always coincide with x-, y-, and z-directions in nature. Moreover, it is also unknown whether Euler angles are spatially variable or not. These assumptions on anisotropic axes limits the usefulness and reliability of the inversion method when applied to field data. If this paper tries to add important contribution to Kong et al. (2022), I would suggest to treat Euler angles as free parameters (extension of inversion theory is needed). This also enhances the importance of the data space inversion because the number of free parameters can be twice as large. I know it is not easy to do so in general, but there is a room for reducing the difficulty of the problem. Kong et al. (2022) pointed out that the vertical conductivity is poorly resolved in general (this paper also shows a similar result). So, I would suggest the authors to examine the effect of inclusion/exclusion of the vertical conductivity, by comparing inversion results with and without the vertical conductivity for the same synthetic model. The latter test can be realized by assuming the vertical conductivity to be (for example) that of background isotropic medium, or the mean of two horizontal values. This treatment reduces the number of parameters by 2/3. Further, such a formulation enables the authors to examine azimuthal anisotropy by letting a rotation angle around the vertical axis be a free parameter without changing the total number of parameters. I think such an attempt is much more interesting than just repeating what former studies have done.
  • Generally speaking, lateral variation of anomalies is resolved by an array observation if the anomaly scale is larger than the site spacing. If the numerical model includes discretized cells whose horizontal size is much smaller than the site spacing, we can say such a discretization is redundant (such a fine discretization is sufficient but not necessary). Unfortunately it is not clear (although quite likely) whether this paper deals with redundant models or not, as only total numbers of parameters are shown for different models. For readers’ understanding, it is helpful to show numbers of discretization in x-, y- and z-directions more explicitly (like Fig. 4 of Kong et al. (2022)). It is nice that data space inversion enables us to handle a larger model than model space inversion, but the number of model parameters does not have to be unnecessarily large. Considering that the roughness is calculated by 20 surrounding cells (line 183), the inverted model parameters are suggested to have large correlation length and therefore the model is likely to be redundant. There are several statistical criteria to test the appropriateness of the number of model parameters for the given data (e.g. AIC). I would suggest the authors to perform such a test and consider optimizing the inversion problem and show the result. As a result, it may be indicated that the data space inversion is not more efficient than model space inversion for the case treated in the present version (anisotropy fixed to x-, y- and z-directions). However it will definitely be more relevant for the cases where Euler angles are also inverted.
  • Kong et al. (2022) uses the RMS (root-mean-squared) misfit. The meaning of the RMS misfit is clear, as it is the , where is the data misfit functional defined by Eq. (5) and is the dimension of . It is preferable to use the RMS misfit, as it’s meaning is clear (it becomes 1 or smaller if calculated data explains observation data within the range of observation errors). However, the misfit used in this paper is differently defined, considering the small values (<0.02). A clear definition is needed. Also, the authors must note that a solution with smaller misfit does not always mean a better solution. Small misfit may be due to overfitting the data, because the synthetic data include 2 % Gaussian noise. This is why a statistical test is important (See previous comment). Any regularized inversion has a trade-off between data fit and model roughness (regularization). A clear presentation is required how the ‘best inversion result’ was chosen.

Minor comments

  • Line 132-134. Generally speaking, any regularized inversion has a trade-off between the data misfit and regularization terms, and so far as I know, there is no general answer how to choose appropriate regularization parameters. However, it seems that the appropriate regularization parameter were uniquely determined by using Eqs. (21) and (22). Does this mean this paper was successful in giving a general answer? No. I find that Eq. (22) includes two unknown parameters, and . The former is a real positive number between 0 and 1 and the latter a positive integer. A clear description is needed how to determine these two parameters and hopefully why they can be determined uniquely.
  • Line 153-154. The total number of synthetic data should be 16,224 instead of 8112, considering each impedance element is a complex quantity.
  • (5) includes observation error (the standard ‘error’ of observations is correct terminology, not ‘deviation’), but it is not shown how the values are given in practice of the synthetic study. Kong et al. (2022) assigned 5% of . I wonder whether the present study did the same. Please clarify.
  • Scaling of data: It is written in line 86 that the model parameter is conductivity value of each discretized cell scaled in logarithmic scale. However, I can not find how the synthetic observed data are scaled, either in linear or in logarithmic scale. Note that the synthetic observation error has to be scaled according to the data scaling so that the RMS misfit has a correct meaning.
  • In the present case, only one case of data noise (2% Gaussian) is examined. The same in Kong et al. (2022). I think it is interesting to examine cases of other noise levels and compare them. Such an examination is not attempted by Kong et al. (2022) and therefore can be a new contribution by this paper.
  • Figure 7. Phase values are presented in radians but the scaling of impedance amplitude is not shown. Do the authors scale it by natural log? If so, I do not understand why different color scales were used for amplitude and phase. If Gaussian noise is given to complex impedance in log scale, resulting variabilities of residuals in log impedance amplitude and phase have to be similar order. Did the authors assumed linear Gaussian noise but scaled the impedance in natural log? It is possible, but I do not understand why. Very confusing.
  • In Figures 4 and 10, ‘normalized misfit’ is given in the vertical axis. It is not clear whether this is the same as ‘misfit’ appears in the text. If this quantity is defined by Eq. 5, I do not understand why it has so small value. Also, ‘noise floor’ is indicated by dashed line in these figures, but I do not understand why the misfit is directly related to the given random noise. If the observation error is given as 5% of , it will be difficult to get such a small misfit. Anyway such a comment is meaningless because the observation error is unknown. Very confusing.
  • Line 183. Only one case of model roughness is examined using 20 surrounding elements. How this number is determined?
  • Line 189-194. Contents of this paragraph is doubtful. I can not see any artefacts only around the anisotropic conductive body in Figure 3. I do not understand how the authors can say that the shape and location of the anisotropic resistive body is better recovered than those of the conductive body. A quantitative discussion is needed here.

Author Response

 please see attached PDF file.

Reviewer 3 Report

Dear Authors

The MS is new and I thank the authors for their valuable contribution. However, I have some comments that need some explanation from the authors.

  • Page 3, Line 92: Different forms can be used for W. Explain why using the1/standard deviation.
  • Page 3 line 98: Why using GN method? It is well known that GN has many disadvantages. In principle, the approximate matrix H used is required to be invertible (and positive definite), but the actual calculation is only positive semi-definite. The required H may be a singular matrix or ill-conditioned. The incremental stability is poor, resulting in non-convergence of the algorithm. Even assuming that the required H is a non-singular matrix, it is not ill-conditioned. If the step size is too large, the local approximation will be incorrect. GN may not converge or even diverge. Also, it belongs to the liner search category: fixed direction, search without step length.
  • I think that submitting a methodology without a field example is a disadvantage. The algorithm needs to be applied to a real field example to convince the reader with the importance and performance of it. Please add one or two field examples.
  •  
  • Regards

Author Response

(1) Page 3, Line 92: Different forms can be used for W. Explain why using the 1/standard deviation.

Reply: The chosen forms of W can reduce the weight of data with poor quality and normalize the data of different orders of magnitude.

(2) Page 3 line 98: Why using GN method? It is well known that GN has many disadvantages. In principle, the approximate matrix H used is required to be invertible (and positive definite), but the actual calculation is only positive semi-definite. The required H may be a singular matrix or ill-conditioned. The incremental stability is poor, resulting in non-convergence of the algorithm. Even assuming that the required H is a non-singular matrix, it is not ill-conditioned. If the step size is too large, the local approximation will be incorrect. GN may not converge or even diverge. Also, it belongs to the liner search category: fixed direction, search without step length.

Reply: Although the GN method has some disadvantages, it shows approximately quadratic convergence behavior. The positive semi-definite of required H can be overcome by adding a large positive number to the diagonal of H, refer to ‘Tomasi and Bro (2012), Multilinear Models: Iterative Methods’. And the data-space method is not strong affected by the singularity and ill-conditioned property of required H. In addition, the step size is no longer fixed in the GN method to stabilize the convergence of the algorithm, and the search strategy is generally used to obtain the appropriate step size, refer to ‘Grayver et al. (2013), Three-dimensional parallel distributed inversion of CSEM data using a direct forward solver, Geophys. J. Int.’.

(3) I think that submitting a methodology without a field example is a disadvantage. The algorithm needs to be applied to a real field example to convince the reader with the importance and performance of it. Please add one or two field examples.

Reply: We have not collected suitable magnetotelluric anisotropic field data and it is challenging to find field data and do inversion at this moment. However, this will be addressed in our future research.

Round 2

Author Response

please see attached PDF file. 

Reviewer 3 Report

Dear Authors

Although the authors replied on my previous comment, but a little work is added to the manuscript and, in general it is not improved. The authors comments should be implemented in the text to improve the text. I still have some criticism as in the following:

  • Please discuss the different forms of W. From your reply, the present form is stable against noise in data. Please prove numerically (by comparing with other forms) the proper choice of W (authors can refer to Menke in this issue) and include this study within the text.
  • I still have major concern concerning the ill-condition of H matrix. As the author replied”

the data-space method is not strong affected by the singularity and ill-conditioned property of required H” please prove that numerically to the reader.

  • Did the authors added a large positive number to the diagonal of H to overcome the positive semi-definite of required H?
  • The incremental stability in the GN is poor, resulting in non-convergence of the algorithm. Please prove that your approach is convergent by trying some experiments to show the reader the efficiency of the technique, otherwise it is not applicable.

Author Response

please see  attached PDF file

Round 3

Reviewer 3 Report

thank you

Author Response

 The English is further polished in this revision.